# The Synergistic Effect of GH13 and GH57 GBEs of *Petrotoga mobilis* Results in α-Glucan Molecules with a Higher Branch Density

**DOI:** 10.3390/polym15234603

**Published:** 2023-12-02

**Authors:** Hilda Hubertha Maria Bax, Aline Lucie Gaenssle, Marc Jos Elise Cornelis van der Maarel, Edita Jurak

**Affiliations:** Bioproduct Engineering, Engineering and Technology Institute Groningen, University of Groningen, Nijenborgh 4, 9747 AG Groningen, The Netherlands; h.h.m.bax@rug.nl (H.H.M.B.); algaenssle@gmx.at (A.L.G.); m.j.e.c.van.der.maarel@rug.nl (M.J.E.C.v.d.M.)

**Keywords:** glycogen branching enzyme, alpha-1,4-transglycosylation, branching activity

## Abstract

Glycogen is a biopolymer consisting of glycosyl units, with a linear backbone connected by α-1,4-linkages and branches attached via α-1,6-linkages. In microorganisms, glycogen synthesis involves multiple enzymes, with glycogen branching enzymes (GBEs) being vital for creating α-1,6-linkages. GBEs exist in two families: glycoside hydrolase (GH) 13 and GH57. Some organisms possess either a single GH13 or GH57 GBE, while others, such as *Petrotoga mobilis*, have both types of GBEs. In this study, the simultaneous use of a GH13 and GH57 GBE each from *Petrotoga mobilis* for α-glucan modification was investigated using a linear maltodextrin substrate with a degree of polymerization of 18 (DP18). The products from modifications by one or both GBEs in various combinations were analyzed and demonstrated a synergistic effect when both enzymes were combined, leading to a higher branch density in the glycogen structure. In this cooperative process, PmGBE13 was responsible for creating longer branches, whereas PmGBE57 hydrolyzed these branches, resulting in shorter lengths. The combined action of the two enzymes significantly increased the number of branched chains compared to when they acted individually. The results of this study therefore give insight into the role of PmGBE13 and PmGBE57 in glycogen synthesis, and show the potential use of both enzymes in a two-step modification to create an α-glucan structure with short branches at a high branch density.

## 1. Introduction

Glycogen, an α-glucan polymer found in various microorganisms, is synthesized in the presence of a carbon source and in response to nutrient limitation [1,2,3], especially during nitrogen deficiency [4,5]. The amount of glycogen accumulated in bacteria can vary from 0.3 to 40% of the cell’s dry weight, with some exceptions reaching as high as 70% [6,7,8,9]. 

Glycogen is a branched polymer composed of glycosyl units. Its structure consists of an α-1,4-linked backbone and branches attached by α-1,6-glycosidic linkages [10,11]. 

Different sources of glycogen also exhibit distinct side chain distribution patterns. The average chain length (ACL) is a commonly used parameter to describe the side chain pattern, which is determined by dividing the number of glucosyl residues by the number of branching points. In glycogen structures with similar molecular weights, a shorter ACL results in a higher percentage of branching, and vice versa [12]. The ACL of glycogen varies considerably among different species. For example, *Bacillus stearothermophilus* has an ACL of 21, while *Aerobacter aerogenes* and *E. coli* have ACLs of 13 and 12, respectively [13,14]. In comparison, *Arthrobacter* and *Mycobacterium* both have an ACL of seven [15]. The ACL and percentage of α-1,6-glycosidic linkages in glycogen play a crucial role in determining its degradation rate, as they influence the interaction with digestive enzymes. The ACL has a significant impact on the activity of glycogen debranching enzymes as it has been suggested that glycogen with a shorter ACL is more difficult to degrade [12,13,16]. 

Bacteria equipped with glycogen that is harder to break down have been found to be capable of sustaining themselves for longer periods during nutrient scarcity, giving them a survival advantage in adverse conditions. On the other hand, bacteria storing glycogen that can be more readily utilized exhaust their energy reserves more rapidly, making them more susceptible to adverse environmental conditions and, consequently, less viable under starvation conditions [12]. 

Several enzymes are critical in glycogen synthesis, including ADP-glucose pyrophosphorylase, glycogen synthase, and glycogen branching enzymes (GBEs) [4,17]. Among these enzymes, GBEs (EC 2.4.1.18) play a crucial role in creating α-1,6-linkages or branch points, thereby defining the ultimate glycogen structure. GBEs are categorized into two families: glycoside hydrolase (GH) family 13 or GH57, based on their primary amino acid sequence and conserved motifs [18,19,20]. Most bacteria typically possess either a GH13 or a GH57 GBE, but certain bacteria harbor both encoding genes, one for GH13 and one for GH57 [19,21]. An example of such a bacterium is *Mycobacterium tuberculosis*, which is associated with tuberculosis [22,23]. Deletion of its GH13 GBE gene (Rv1326c) resulted in non-viable mutants [24], while its GH57 GBE (Rv3031) has first been proposed to be involved in lipopolysaccharide synthesis [25,26] and later to be a glucosylglycerate branching enzyme participating in the biosynthesis of methylglucose lipopolysaccharides [27]. 

In the genomes of most members of the *Petrotogaceae* family, two GBE genes are found, one encoding a GH13 GBE and the other a GH57 GBE [21]. A study conducted by Zhang et al. demonstrated that both of these GBE genes from *P. mobilis* are functional GBEs in vitro. The GH13 GBE from *P. mobilis* (PmGBE13) exhibited high activity on amylose and synthesized highly branched α-glucans with molecular weights ranging from 10^6^ to 10^7^ Da, suggesting an essential role in glycogen synthesis [21]. On the other hand, the GH57 GBE from *P. mobilis* showed very low branching activity on amylose, synthesizing products of lower molecular weight (~10^4^ Da) and raising uncertainty about its direct involvement in generating the typical highly branched glycogen structure. Therefore, further research is needed to elucidate the precise role and significance of the GH57 GBE in glycogen biosynthesis and understand the functions of both GBEs in the glycogen metabolism of these microorganisms.

Furthermore, GBEs are enzymes that are often used in the food industry to produce modified starches with a low molecular weight and a high degree of branching [28]. 

The aim of the present study was to characterize the activity of both the GH13 and GH57 GBE from *P. mobilis* together. The hypothesis was that both GBEs in *P. mobilis* play a role in glycogen synthesis and have a synergistic effect in creating an α-glucan structure with short branches at a high branch density. To investigate this, a highly defined linear substrate (maltooctadecaose; MD18) was used, allowing for a detailed examination of the catalytic mechanisms of both enzymes. It is suggested that the α-1,4-transglycosylation activity of GH57 GBEs elongates linear glucan chains or branches, making them available for GH13 GBEs to create new branches [29]. Additionally, GH57 GBEs can hydrolyze branches formed by GH13 GBEs, resulting in shorter branches and a higher branch density. The combined action of GH13 and GH57 GBEs is proposed to be crucial for *P. mobilis* in producing a glycogen structure with a short ACL and high branch density. This synthesized glycogen structure is difficult to degrade, making it an efficient and durable energy storage molecule for *P. mobilis*. This characteristic enables *P. mobilis* to survive for extended periods, particularly under conditions of nutrient limitation, providing the microorganism with a distinct survival advantage. The results of this study shed light on the cooperative role of both GH13 and GH57 GBEs in glycogen synthesis, contributing to a better understanding of glycogen metabolism and energy storage in *P. mobilis*. Additionally, the study shows the potential use of the synergistic effect between GH13 and GH57 GBEs in starch modification and its application as a functional food ingredient.

## 2. Materials and Methods

### 2.1. Materials and Commercial Enzymes

Maltodextrin MD18 (Maltooctadecaose) was obtained from CarboExpert (Yuseong-gu, Daejeon, Republic of Korea), and linear maltodextrin MD7 (Maltoheptaose) was purchased from Sigma-Aldrich (Saint Louis, MO, USA). The two debranching enzymes, isoamylase from *Pseudomonas* sp. (E-ISAMY, 200 U/mL) and pullulanase M1 from *Klebsiella planticola* (E-PULKP, 650 U/mL), were both purchased from Megazyme (Bray, Ireland), and MagicMedia and HisPur^TM^ Ni-NTA Resin from ThermoFischer Scientific (Waltham, MA, USA). All other materials were of analytical grade or higher.

### 2.2. Enzyme Production and Purification

The gene encoding the GH 13 GBE from *Petrotoga mobilis* SJ95 (PmGBE13, Genbank: ABX32021.1)*,* and the gene encoding the GH57 GBE from *Petrotoga mobilis* SJ95 (PmGBE57, Genbank: CP000879.1) were codon optimized by GenScript, cloned into a pET28a(+) vector which contained a C-terminal His-Tag (GenScript USA Inc., Piscataway, NY, USA) and were expressed in *E. coli* BL21(DE3). Single colonies grown on agar plates containing kanamycin were used for starter cultures for each strain. These cultures were grown in LB media (Luria-Bertani media, 1% NaCl, 1% tryptone, 0.5% yeast extract) with 50 µg/mL kanamycin at 37 °C and 150 rpm for 18 h. Afterwards, an aliquot (25 mL) was used for a main culture (500 mL, MagicMedia^TM^ with 50 µg/mL ampicillin or kanamycin), which was grown at 30 °C and 150 rpm for 25 h.

The cells were obtained by centrifugation (5000× *g*, 10 min, 4 °C) and lysed by sonication while kept on ice (10 min with rounds of 30 s on and 30 s off; amplitude 20%, pulse 50%) in lysis buffer (20 mM sodium phosphate, pH 7.4, 500 mM NaCl, 20 mM Imidazole). The cell debris was then harvested by centrifugation (12,000× *g*, 15 min, 4 °C) and the extracts were purified with an ÄKTA system (GE Healthcare, Chicago, IL, USA) by using a 5 mL HisPur^TM^ Ni-NTA column and a flow rate of 1.0 mL/min. First, the column needed equilibration with eluent A (20 mM sodium phosphate buffer, pH 7.4, 500 mM NaCl, 20 mM Imidazole). Then, the samples were loaded onto the column. The column was washed until the UV signal (280 nm) stabilized. Proteins attached to the column were eluted with eluents A and B (20 mM sodium phosphate, pH 7.4, 500 mM NaCl, 500 mM Imidazole) with the following gradient profile: 0–10 min (0–50% B), 10–20 min (50% B), 20–30 min (50–100% B), and 30–40 min (100% B). The fractions which contained the proteins of interest were collected, and subsequently desalted with buffer exchange (20 mM sodium phosphate, pH 7.4). Concentration was performed using an Amicon^®^ Ultra filter (30,000 MWCO, 15 mL) from Sigma-Aldrich (Saint Louis, MO, USA). The protein concentrations were determined with the Pierce BCA Protein Assay (Thermo Fisher Scientific Inc., Waltham, MA, USA) and purity was verified by SDS-PAGE (Appendix A). The purified proteins were stored at −80 °C.

The activity on potato amylose V (AVEBE, Veendam, The Netherlands) was tested using the iodine assay [30]. Different concentrations (25–300 µg/mL) of GBEs were incubated with 1 mg/mL potato amylose in 50 mM sodium phosphate buffer, pH 7.5 at 50 °C [21]. 15 µL aliquots were taken from 10 min to 2 h and mixed with 100 µL iodine reagent (0.26% KI, 0.026% I^2^, 5 mM HCl). After all samples had been collected, the absorbance of the iodine–amylose complex was measured at 610 nm with a spectrophotometer (SpectraMax Plus 384 Microplate Reader, Molecular Devices, Sunnyvale, CA, USA). A decrease in absorbance of 1.0 per min at 610 nm was defined as one unit of activity.

### 2.3. Enzyme Reactions and Analysis of Activity with Reducing End Assay

The branching and non-branching activity of the GBEs was determined on potato amylose (2.5 mg/mL) in 10 mM sodium phosphate buffer at 50 °C, gently rotating head-over tail, with an enzyme dose of 10 and 20 mg/g substrate for PmGBE13 and PmGBE57, respectively. Samples were incubated for 0.5, 1, 1.5, 2, and 2.5 h before stopping the reaction by boiling for 5 min. 

Debranching of the GBE-modified glucans was conducted by diluting the samples twice in sodium acetate buffer, pH 4.5, and treating them with isoamylase (1 U/mg substrate) and pullulanase (0.7 U/mg substrate) for 24 h at 40 °C. 

The amount of reducing ends in all branched and debranched samples was analyzed with the pAHBAH (4-hydroxybenzoic acid hydrazide) assay. Samples of 50 µL containing 2.5 mg/mL (branched samples) and 1.0 mg/mL (debranched) amylose were incubated with 200 μL pAHBAH solution (containing 1/5 of 5% pAHBAH in 0.5 M HCl and 4/5 of 0.5 M NaOH) for 30 min at 70 °C and the absorbance was measured at 490 nm using a spectrophotometer with D-glucose as the standard. From the reducing ends of the branched product, the non-branching activity was determined. The branching activity was calculated from the reducing ends of the debranched product minus the branched product. One unit of activity was defined as 1 µmol reducing ends released or transferred per minute. 

The definition of activity U^B^ (units-branching activity) allowed a fair comparison of the activity of GBEs, since it excluded the products from hydrolytic and elongation activity. The increase in reducing ends from branched to debranched products represented the branching activity, U^B^, defined as;
(1)$$U^{B}=\frac{\Delta R\;after\;debranching\;\left[\mu mol\right]-\Delta R\;before\;debranching\;\left[\mu mol\right]}{\Delta time\;\left[min\right]}$$

One U^B^ of activity was defined as 1 μmol reducing ends transformed to branching point per minute.

### 2.4. Chain Length Distribution with Anion Exchange Chromatography

To elucidate the mode of action of the GBEs, an incubation was performed on MD18 (0.5 mg/mL) and with an enzyme dose of 1 U^B^/g substrate (U^B^, unit of branching activity on MD18). After this incubation, both the branched and debranched samples were analyzed with High-Performance Anion Exchange Chromatography coupled with Pulsed Amperometric-EDet1 Detection (HPAEC-PAD) with the Gold Standard PAD waveform, using a Dionex ICS-6000 system with a CarboPac^TM^ PA100 column (ThermoFischer Scientific; Waltham, MA, USA). Samples were analyzed at a concentration of 0.1 mg/mL maltodextrins, with an injection volume of 10 µL and a flow rate of 0.25 mL/min. Eluents A (0.1 M NaOH) and B (0.1 M NaOH, 1 M NaOAc) were used to elute the samples, with a gradient profile as follows: 0–50 min (5–40% B), 50–65 min (40–100% B), 65–70 min (100% B) and a re-equilibration step at 5% B. Chromeleon software version 7.2.9 was used to estimate the peak areas of representative chromatograms (Appendix A) and the increase in linear chains was estimated as Δ peak area (untreated—treated for 24 h). With the increase in peak area after debranching of 24 h modified samples minus the increase in peak area after debranching of untreated substrate, the Δ peak area of branched chains was calculated.

### 2.5. Statistical Analysis

All experiments were conducted in triplicates and the data was presented as means with standard deviations. The statistical analysis was conducted with one-way analysis of variance (ANOVA) and a post-hoc Bonferroni test (95% confidence interval) using Stata17 (StataCorp., College Station, TX, USA). 

## 3. Results and Discussion

### 3.1. Activity of Glycogen Branching Enzymes on Linear Maltodextrin

The enzymatic activity of two glycogen branching enzymes (GBEs) from *Petrotoga mobilis*, specifically a GH13 and a GH57 GBE, was investigated using linear maltooctadecaose (MD18) as a substrate. The results showed that PmGBE13 exhibited significantly higher activity compared to PmGBE57 (as shown in Table 1). Specifically, the branching activity of PmGBE13 was found to be 32 times higher than that of PmGBE57, while the non-branching activity was only 7.4 times higher. Consequently, the ratio of branching to non-branching activity differed significantly between the two enzymes, with PmGBE13 displaying a significantly higher B:NB ratio (3.7 ± 0.3) as opposed to PmGBE57 (0.8 ± 0.0). The ratio of 0.8 ± 0.0 for PmGBE57 suggests that this enzyme possesses equal branching and non-branching activity. 

Comparing these findings to previous studies on other GBEs, it appears that GH13 GBEs typically have higher branching activity, leading to a degree of branching in the range of 7–13% [21,31,32]. Conversely, GH57 GBEs generally exhibit relatively lower branching activity and result in degrees of branching ranging from 3% to 8.5% [21,31,33,34]. The observed difference in activity between PmGBE13 and PmGBE57 indicates that both enzymes likely serve distinct roles in the glycogen synthesis pathway of *Petrotoga mobilis*.

### 3.2. Chain Length Distribution of Modified Maltodextrin DP18 with a Combination of GH13 and GH57 Glycogen Branching Enzymes

The linear maltodextrin MD18 was used as a substrate to facilitate a detailed analysis of the changes in chain length distribution. The chain length distribution of MD18 was examined after individual modification with PmGBE13 and PmGBE57, as well as simultaneous modification with both enzymes.

When MD18 was subjected to PmGBE13, nearly all the DP18 molecules were consumed within a 24 h period, leading to the formation of shorter linear chains and branched chains (Figure 1). The linear chains were mostly in the range from DP8 to DP12, whereas the branches were shorter (from DP5 to DP8). A study conducted by Zhang et al. demonstrated that PmGBE13, when acting on amylose as a substrate, produced side chains ranging from DP2 to DP15 [21]. In contrast, when PmGBE57 acted on amylose, it generated considerably shorter side chains ranging from DP3 to DP5 [21]. Figure 1 shows that PmGBE57 created primarily linear chains, although some branches with lengths of DP16 and DP17 were also detected. Notably, PmGBE57 demonstrated the ability to elongate linear chains, even surpassing the length of the original DP18 substrate. The alpha-1,4-transferase activity of GH57 GBEs has been previously described in GBEs, such as *Thermotoga maritima* GBE57, *Thermococcus kodakarensis* GBE57, and *Meiothermus* sp. GBE57 [35].

When the MD18 substrate was subjected to a one-pot reaction involving both PmGBE13 and PmGBE57, the chain length distribution pattern observed after 24 h closely resembled the pattern obtained from the modification with only PmGBE13. However, the linear chains produced in the presence of PmGBE57 were slightly shorter. This outcome can be attributed to the relatively high hydrolytic activity exhibited by PmGBE57. The hydrolytic activity of an enzyme involves the breaking down of molecules, and in this case, it led to the generation of shorter linear chains during the reaction involving both enzymes, contributing to the observed difference in chain lengths compared to the reaction with PmGBE13 alone.

### 3.3. Two-Step Modification of Linear Maltodextrin DP18 with GH13 and GH57 Glycogen Branching Enzymes

It remains uncertain whether the GH13 and GH57 GBE in *Petrotoga mobilis* are expressed simultaneously during in vivo glycogen synthesis. To investigate the impact of a two-step modification on the chain length distribution, additional experiments were conducted. Figure 2 presents the distribution of linear and branched chains after a two-step reaction, involving either first PmGBE13 followed by PmGBE57, or vice versa. When the substrate underwent a 24 h modification with PmGBE13 as the first step, followed by PmGBE57 in the second step, it resulted in the creation of mainly linear chains falling within the DP8–DP12 range. This distribution was similar to what was observed in the presence of only PmGBE13 or in the one-pot reaction combining both PmGBE13 and PmGBE57. 

However, a notable difference was observed in the length of the branched chains. The two-step reaction, beginning with PmGBE13 followed by PmGBE57, produced shorter branches. The majority of the branches had lengths ranging from DP2 to DP5 (74%), with a smaller fraction ranging from DP6 to DP10 (25%) (Table 2). In contrast, when only PmGBE13 was involved in the modification (as shown in Figure 1), the branches were primarily in the DP6–DP10 range (59%), with a minor quantity falling in the DP11–DP18 range (12%) (Table 2). The activity of PmGBE57 in the secondary step appeared to lead to the hydrolysis of the branches created during the first step, which were the result of PmGBE13’s branching activity. Interestingly, PmGBE57 demonstrated a specific hydrolysis of the branches, with only certain linear chains of DP8, DP11, and DP12 being affected. The ACL of the branched chains after the secondary modification with PmGBE57 decreased from 7.1 to 4.3 (Table 2). 

When performing a two-step modification with first PmGBE57 and then PmGBE13, the resulting product exhibited significant differences compared to the other scenarios. As shown in Figure 1, only PmGBE57 led to the hydrolysis of shorter linear chains, and these hydrolyzed chains served as the starting product for PmGBE13 in the secondary step. However, Figure 2 illustrates that only a minor amount of linear and branched chains were formed by PmGBE13. This observation can be explained by the minimum substrate length required for PmGBE13 to function effectively. PmGBE13 necessitates a donor substrate of at least DP13 in length and cannot utilize shorter chains for either hydrolysis or branching purposes [21].

Overall, as seen from the peak areas in Figure 1 and Figure 2, both the modification with only PmGBE57 and the combination of first PmGBE57 and then PmGBE13 resulted in a product with a substantially low number of branched chains. The modifications with only PmGBE13 or the combination of both enzymes did not show clear differences in the amount of branched chains produced. However, a two-step modification with first PmGBE13 and then PmGBE57 led to an increased number of branched chains. This suggests that the order in which the enzymes are applied in the two-step reaction has a significant impact on the final chain length distribution and branching patterns.

The total number of branched chains and the ACL of α-glucans after modification with GBEs in different combinations differed significantly, and therefore also resulted in structures with a different branch density. Figure 3 shows a simplified model of the α-glucan structures based on the results of the chain length distribution (Table 2). PmGBE57 modification alone resulted in an α-glucan with relatively long branches and a low branch density, whereas the combination of PmGBE13 and PmGBE57 in a one-step reaction resulted in an α-glucan structurally very similar to only PmGBE13 modification. A two-step modification with both PmGBE13 and PmGBE57 showed a product with short branches and an increased branch density.

The increased branch density and shorter branch length are likely to contribute to the glycogen’s resistance towards digestion by glycogen-degrading enzymes. The glycogen structure synthesized by *Petrotoga mobilis*, characterized by higher branch density, enables the organism to endure harsh conditions for extended periods. This type of glycogen structure, produced through the collaboration of GBEs from families GH13 and GH57, serves as enduring energy source, providing crucial benefits for the organism’s survival. From an industrial perspective, the decreased digestibility of α-glucans produced with this combination of enzymes is an interesting characteristic for applications as functional food ingredients or in other industries where a resistance towards digestive enzymes is favorable [28]. 

### 3.4. Modification of Maltodextrin DP7 with GH13 and GH57 Glycogen Branching Enzymes

Both PmGBE13 and PmGBE57 display distinct substrate preferences, and their enzymatic activities also vary depending on the substrate type. For instance, the activity of both PmGBE13 and PmGBE57 is considerably lower on MD18 compared to amylose, as previously reported [21]. To investigate their activities, a shorter linear substrate called maltoheptaose (MD7) was used for analysis. Figure 4 demonstrates that PmGBE13 was unable to utilize this shorter linear substrate, which aligned with the minimum substrate length of DP13 reported by Zhang et al. [21]. Conversely, PmGBE57 showed the ability to utilize the short DP7 chains, as well as the DP6 chains present in the starting substrate. This resulted in the production of predominantly shorter linear chains of DP4 and glucose, with no branching observed. Despite previous reports suggesting a minimum substrate length of DP17 for PmGBE57 [21], Figure 4 illustrates that the enzyme could utilize shorter chains but only for hydrolytic activity. When PmGBE13 and PmGBE57 were combined in a one-pot reaction, hydrolyzed linear chains in the DP4–DP6 range were generated. However, compared to using only PmGBE57, a smaller portion of the initial DP7 was utilized in the combination reaction due to the lack of activity of PmGBE13 on this short substrate. These results corroborate the observations in Figure 2, where the activity of PmGBE13 on an already hydrolyzed product of PmGBE57 modification of MD18 was low.

## 4. Conclusions

*Petrotoga mobilis* possesses two distinct glycogen branching enzymes (GBEs), named PmGBE13 and PmGBE57. PmGBE13 is substantially more active than PmGBE57, with 32 times more branching and 7.4 times more non-branching activity.

Incubation of MD18 with either only PmGBE13 or a one-pot reaction with both PmGBE13 and PmGBE57 resulted in similar products with linear chains ranging from DP8 to DP 12 and branches from DP5 to DP8. The products obtained from a two-step modification with first PmGBE13 and then with PmGBE57 gave a similar distribution of linear chains. However, the majority of the branches were shorter (DP2–DP5), which were in line with the product obtained with only PmGBE57. Modification with PmGBE47 followed by reaction with PmGBE13 resulted in very low amounts of new chains and branches, likely due to PmGBE57 hydrolyzing the substrate to chains shorter than the minimum chain length of PmGBE13 (DP13). This theory was further supported by the inability of PmGBE13 to utilize MD7 (composed of DP6 and DP7). PmGBE57, on the other hand, hydrolyzed the substrate to DP4 and glucose. Overall, the study’s findings reveal a synergistic effect when both enzymes are combined, leading to a higher branch density in the glycogen structure. In this combined action, PmGBE13 initiates the formation of longer branches, while PmGBE57 subsequently hydrolyzes these branches to a shorter length. This cooperative action of the two enzymes results in a significantly higher number of branched chains compared to when they act individually. 

## Figures and Tables

**Figure 1 polymers-15-04603-f001:**
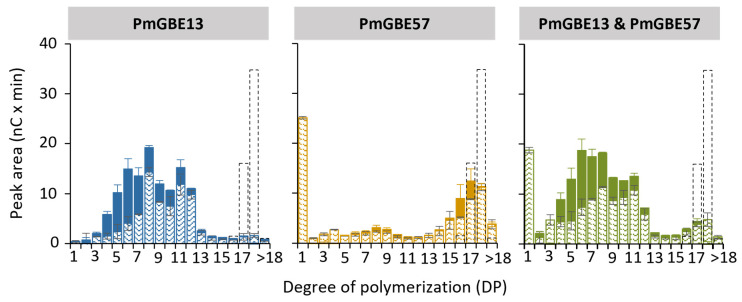
Peak areas of MD18 in linear (pattern fill) and branched (solid fill) chains after treatment with PmGBE13 (blue), PmGBE57 (orange) or PmGBE13 and PmGBE57 (green) (*Petrotoga mobilis*) at 1 U^B^/g S for 24 h compared to the untreated substrate (dotted bars). The peak area of branched chains was calculated by the increase in peak area after debranching of 24 h GBE-modified samples minus the peak area of the 24 h GBE-modified samples before debranching.

**Figure 2 polymers-15-04603-f002:**
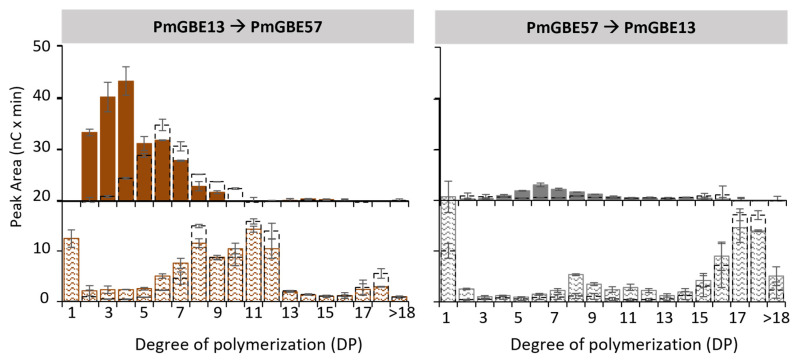
Peak areas of MD18 in linear (pattern fill) and branched (solid fill) chains after treatment with PmGBE13 and PmGBE57 (*Petrotoga mobilis*) in two steps at 1 U^B^/g S for 24 h compared to the substrate treated after only one step (dotted bars). The peak area of branched chains was calculated by the increase in peak area after debranching of 24 h GBE-modified samples minus the peak area of the 24 h GBE-modified samples before debranching.

**Figure 3 polymers-15-04603-f003:**
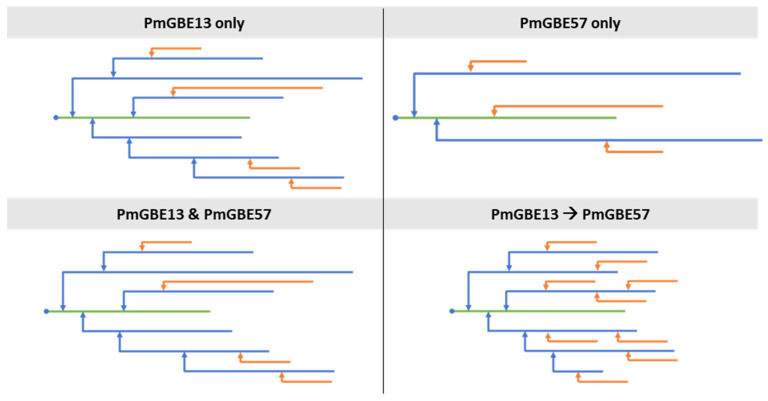
Simplified models of α-glucan structures after modification with PmGBE13 only, PmGBE57 only, PmGBE13 and PmGBE57 in a one-step modification or PmGBE13 and PmGBE57 in a two-step modification, based on the results obtained on chain length distribution by HPAEC. Blue dots represent the reducing end of each α-glucan molecule, green lines indicate the linear backbone chains carrying the reducing end, blue lines show long substituted branches, and orange lines show external branches.

**Figure 4 polymers-15-04603-f004:**
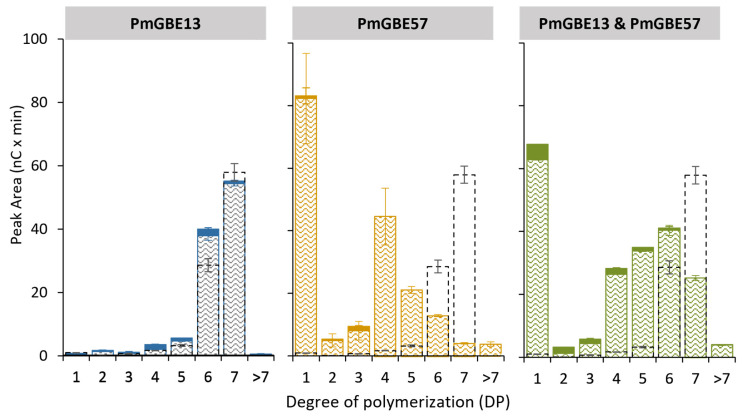
Peak areas of MD7 in linear (pattern fill) and branched chains (solid fill) after treatment with PmGBE13, PmGBE57, or PmGBE13 and PmGBE57 (*Petrotoga mobilis*) at 1 U^B^/g S for 24 h compared to the untreated substrate (dotted lines). The peak area of branched chains is calculated by the increase in peak area after debranching of 24 h GBE-modified samples minus the peak area of the 24 h GBE-modified samples before debranching.

**Table 1 polymers-15-04603-t001:** Activity of PmGBE13 and PmGBE57 (*Petrotoga mobilis* glycogen branching enzyme GH13 and GH57) on MD18, analyzed as increase (non-branching activity) or transfer (branching activity) of reducing ends.

	PmGBE13	PmGBE57
Non-branching activity [mU^NB^/mg E]	42.0 ± 1.6	5.7 ± 0.4
Branching activity [mU^B^/mg E]	154.2 ± 15.8	4.8 ± 0.4
Ratio B:NB *	3.7 ± 0.3	0.8 ± 0.0

Average of three independent measurements with standard deviation. * ratio branching activity to non-branching activity.

**Table 2 polymers-15-04603-t002:** Chain length distribution and average chain length (ACL) of PmGBE13 and PmGBE57 (*Petrotoga mobilis* glycogen branching enzyme GH13 and GH57) branched α-glucans from MD18.

		PmGBE13 Only	PmGBE57 Only	PmGBE13 and PmGBE57	PmGBE13 → PmGBE57	PmGBE57 → PmGBE13
Linear chains	DP 2–5 (%)	8.8 ± 4.0 ^a^	39.8 ± 3.6 ^d^	30.0 ± 3.5 ^c^	21.5 ± 0.1 ^b^	27.9 ± 6.0 ^c^
	DP 6–10 (%)	52.1 ± 1.5 ^c^	11.8 ± 0.8 ^a^	40.4 ± 0.8 ^b^	42.7 ± 1.3 ^b^	15.5 ± 0.04 ^a^
	DP 11–18 (%)	38.3 ± 5.7 ^a^	43.9 ± 2.0 ^a^	28.4 ± 3.0 ^a^	34.8 ± 1.1 ^a^	51.5 ± 4.6 ^a^
	DP > 18 (%)	0.8 ± 0.2 ^a^	4.6 ± 0.8 ^b^	1.2 ± 0.3 ^a^	0.9 ± 0.2 ^a^	5.1 ± 1.4 ^b^
	ACL (DP)	9.6 ± 0.5 ^b^	9.7 ± 0.5 ^b^	8.1 ± 0.3 ^a^	8.6 ± 0.03 ^a^	11.1 ± 0.9 ^b^
Branched chains	DP 2–5 (%)	23.3 ± 0.4 ^a^	30.4 ± 10.4 ^a^	29.9 ± 2.2 ^a^	73.8 ± 1.8 ^b^	32.4 ± 2.3 ^a^
	DP 6–10 (%)	59.2 ± 3.2 ^b^	14.8 ± 1.4 ^a^	57.7 ± 1.5 ^b^	25.1 ± 1.7 ^a^	51.6 ± 5.3 ^b^
	DP 11–18 (%)	12.2 ± 3.3 ^b^	54.7 ± 8.8 ^c^	11.8 ± 3.2 ^b^	1.1 ± 0.1 ^a^	14.8 ± 2.1 ^b^
	DP > 18 (%)	0.5 ± 0.1 ^a^	0.2 ± 0.2 ^a^	0.6 ± 0.5 ^a^	0.003 ± 0.003 ^a^	1.1 ± 0.9 ^a^
	ACL (DP)	7.1 ± 0.1 ^b^	12.0 ± 0.1 ^c^	7.0 ± 0.3 ^b^	4.3 ± 0.1 ^a^	8.1 ± 1.2 ^b^

^a–d^ Significantly different groups from other values of the same row.

## Data Availability

Data are contained within the article.

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
