# Peer review of "The Synergistic Effect of GH13 and GH57 GBEs of Petrotoga mobilis Results in α-Glucan Molecules with a Higher Branch Density"

_polymers, 2023, doi:10.3390/polym15234603_

Round 1
Reviewer 1 Report
Comments and Suggestions for Authors
In this paper, this study reveals the role of PmGBE13 and PmGBE57 in glycogen synthesis and shows the potential use of both enzymes in two-step modifications to produce α-glucan structures with short branching and high branching density. Overall, the paper has certain novelty and advantages for this field research work, and has value for publishing in the Journal. I suggest this manuscript can be published after the following minor revisions:
1. Please further distill the introduction, rendering the research background exposition more lucid.
2. It is suggested to add the data and results of the highlights of this experiment in the conclusion section to increase the persuasiveness of the conclusion
3. Line 344-345 The significance analysis of the data in Table 2 needs to be further improved. Why are some data marked while others are not?
Author Response
Reviewer 2
In this paper, this study reveals the role of PmGBE13 and PmGBE57 in glycogen synthesis and shows the potential use of both enzymes in two-step modifications to produce α-glucan structures with short branching and high branching density. Overall, the paper has certain novelty and advantages for this field research work, and has value for publishing in the Journal. I suggest this manuscript can be published after the following minor revisions:
- Please further distill the introduction, rendering the research background exposition more lucid.
We thank the reviewer for the advice and have edited the introduction substantially to make it more lucid.
- It is suggested to add the data and results of the highlights of this experiment in the conclusion section to increase the persuasiveness of the conclusion
We have rewritten the conclusion and added a short summary (with numbers) for each results section
- Line 344-345 The significance analysis of the data in Table 2 needs to be further improved. Why are some data marked while others are not?
The statistical analysis was conducted for each row and only samples with significanly different values were marked with a letter. For more clarity, letters have been added to each value as well as a short explanation on the process was added to the table footnote.

Reviewer 2 Report
Comments and Suggestions for Authors
Bax and her colleagues have chosen an interesting topic as the subject of their research, and in recent years they have published several publications in this area. I see an important methodological problem in this manuscript, which calls into question the authenticity of the results.
It is known that the PAD signal intensity decreases when the degree of polymerization (DP) increases, so the relative composition cannot be given based on area %. This means that the longer oligomers represent smaller fraction compared to their molar ratio than the real one, when the areas are summed up. For correct calculation calibrations are required.
Product of pullulanase reaction may be glucose, maltose or maltotriose. How do you explain that the area of the peaks belonging to mono, di- and trisaccharides did not increase after the debranching reaction based on the figures B in the Supplement?
Minor questions
- Introduction is too long, it should be shortened.
- The referenced articles 10 and 11 are out of date. Anhydroglucopyranose and glucosyl unit are not synonymous, glucosyl is the correct form..
- Abbreviation pAHBAH was not defined.
- The y-axis label is missing in part B of all supplementary figures.
- Which type of isoamylase and pullulanase were used?
- Have the retention time of linear and branched oligomers with same DP equal in HPAEC?
Author Response
Reviewer 1
Bax and her colleagues have chosen an interesting topic as the subject of their research, and in recent years they have published several publications in this area. I see an important methodological problem in this manuscript, which calls into question the authenticity of the results.
It is known that the PAD signal intensity decreases when the degree of polymerization (DP) increases, so the relative composition cannot be given based on area %. This means that the longer oligomers represent smaller fraction compared to their molar ratio than the real one, when the areas are summed up. For correct calculation calibrations are required.
We agree with the reviewer and have removed the Figure 3 but we believe that very general differences in level of produced branches can still be concluded.
Product of pullulanase reaction may be glucose, maltose or maltotriose. How do you explain that the area of the peaks belonging to mono, di- and trisaccharides did not increase after the debranching reaction based on the figures B in the Supplement?
It is our opinion that the lack of observed increase is due to density of branches. It is known that in highly dense/branched structures – pullulanase and iso-amylase can’t access shorter branches which we think is the case here. We have discussed and published this previously in: The influence of amylose content on the modification of starches by glycogen branching enzymes Aline L.O. Gaenssle, Marc J.E.C. van der Maarel, Edita Jurak.
Minor questions
- Introduction is too long, it should be shortened.
We have shortened the introduction and removed all parts not stricly necessary.
- The referenced articles 10 and 11 are out of date. Anhydroglucopyranose and glucosyl unit are not synonymous, glucosyl is the correct form..
The naming has been corrected to glucosyl unit and the referenced articles have been replaced to more current articles.
- Abbreviation pAHBAH was not defined.
We appologize for the unclear description and have updated the description and added pAHBAH to the list of abbreviations.
- The y-axis label is missing in part B of all supplementary figures.
Y axis are the same hight and spacing nC intensity value for A and B so in all the figures, for visual representation and less text, we have opted to leave the numbers out of the figure B but have now added a note in the title that the values of the Y axis are the same for both A and B figure.
- Which type of isoamylase and pullulanase were used?
The following enzymes were used for debranching: Isoamylase from Pseudomonas sp. and pullulanase M1 from Klebsiella planticola as was mentioned in the material section. For futher clarification, the section has been renamed to “Materials and commercial enzymes”
- Have the retention time of linear and branched oligomers with same DP equal in HPAEC?
The retention time for branched oligomers is less than for linear chaims with the same DP and sometimes appear as small peaks inbetween the peaks for linear chains. However, in this study, the branches were only studied after debranching the carbohydrate, making them linear chains as well.

Round 2
Reviewer 2 Report
Comments and Suggestions for Authors
The manuscript of Bax and co-workers has been significantly improved, and the authors have addressed all the questions that arose. Revised version of the manuscript is now suitable for publication in Polymers, I recommend its acceptance.